# In the quest for effective factors of satisfaction with life: Insights from intra-couple interaction and financial management variables

**Monika Baryła-Matejczuk**[1]*, **Wiesław Poleszak**[1], **Kamil Filipek**[2], **Andrzej Cwynar**[3], **Tomasz Żółtak**[4]

**1** Institute of Psychology and Human Sciences, WSEI University, Lublin, Poland, **2** Institute of Sociology, Maria Curie-Skłodowska University, Lublin, Poland, **3** Institute of Public Administration and Business, WSEI University, Lublin, Poland, **4** Institute of Philosophy and Sociology, Polish Academy of Science, Warsaw, Poland

* monika.baryla@wsei.lublin

**Data Availability Statement:** All files are available from the Mendeley Data database: Baryła-Matejczuk, Monika (2022), "Factors of relationship

## Abstract

The aim of the study was to investigate the factors affecting life satisfaction with reference to particular reports from both partners in the relationship. The study was conducted within a group of 500 heterosexual couples. The accuracy of the actor-partner interdependence models (APIM) which offer in-depth insights into the dyadic relationships between female and male partners were estimated. The results of the chi-square test enabled us to reject the hypothesis of actor indistinguishability, therefore the model proposing distinguishability with respect to gender was explored further. The results suggest that women's credit management behavior patterns predict changes in her assessment of well-being. Moreover, the financial behavior patterns of women have an impact on the assessment of well-being as reported by their male partners. Moreover, shared goals and values turned out to be significant with regard to the assessment of quality of life for both women and men. The obtained results provide an insight into the difficulties experienced within relationships and indicate the importance of the roles assumed in various areas of financial management.

## Introduction

Despite the increasing proportion of single person households both in the US and the EU [1,2], committed relationships–particularly marriages–are viewed as a desired social value that advances the interests of individuals and society as a whole. However, available research shows that the benefits of being in a relationship hinge on its quality, which manifests itself in relationship stability and the satisfaction of partners, among others. Poor relationship quality carries risks, and unhappy marriages provide fewer benefits than happy ones [3].

Empirical evidence drawing on Renne's research [4] indicates that marital quality is a significant component of psychosocial well-being; it also has a positive influence on overall happiness and on subjective rating of their own general health status by an individual. It has been

quality/SWLS ", Mendeley Data, V1, doi: 10.17632/
zr8tnvk43m.1.

**Funding:** This research was funded by the Ministry
of Science and Higher Education (https://www.gov.
pl/web/science), Republic of Poland, grant number
0057/DLG/2016/10. The APC was funded by the
University of Economics and Innovation, Lublin,
Poland. The funders had no role in study design,
data collection and analysis, decision to publish, or
preparation of the manuscript.

**Competing interests:** The authors have declared
that no competing interests exist.

shown that poor quality marriages and high levels of marital stress are associated with higher levels of depression and a reduction in overall life satisfaction for both men and women [5]. Relationship satisfaction–a key dimension of relationship quality–has an impact on job satisfaction (cf. [6,7]) and affects the situation of the children [8]. For most married couples in modern societies, the quality of their marriage strongly influences their personal happiness [9]. This also pertains to committed relationships in general. It has been found that the quality of relationship is a significant correlative of overall life satisfaction and experienced well-being [10].

Overall life satisfaction and well-being depends on many factors and one of them is the quality of built interpersonal relationships. Due to the growing material aspirations of individuals getting to know the world from digital sources [11], those related to intra-household financial management are among the most essential [12]. For example, it has been found that "relative income" has a critical impact on individual well-being and couple relationships [13]. In a recent survey on a sample representative for the US adult population, Ramsey Solutions [14] found that money is the number one issue couples argue about. Disagreements and arguments about money have been identified as strong predictors of deterioration in relationship satisfaction [15] divorce [16] and conflict tactics, including increased inclination to engage in heated arguments as opposed to calm discussions [17]. There is a great body of literature documenting how personal finance-related interactions between relationship partners (e.g., communication, disagreements) and their emotions (e.g., financial anxiety, financial strain) link to various aspects of relationship quality (satisfaction, stability, conflict) (see, for instance, [18] for a review).

However, little is known about how such life satisfaction relates to healthy (or unhealthy) financial behaviors of both relationship partners. In this article we use the Couples and Finances Theory (CFT) [19] to explore the connection between satisfaction with life–as the dependent variable–and: (i) financial management behaviors and (ii) couple interaction factors (share goals and values, harsh start up) as the independent variables. Consequently, the following research question is pursued here: Do financial management behaviors of one partner are associated with the perception of life quality by the other?.

We believe that our work contributes to the literature in several ways. First, we respond to the paucity of empirical evidence on the link between financial behaviors of relationship partners and the well-being. Second, we explore this link using dyadic data, which creates an opportunity to compare reports of both partners. Third, our dyadic dataset enabled us to make a cross check: we examined not only how healthy financial behavior on the part of a respondent links to his or her perceived satisfaction with life, but also how it links to the life satisfaction perceived by her or his partner. Fifth, our data has been collected in Poland. Given that the research on the link between intra-household financial matters and well-being comes almost exclusively from the US, we believe that findings from a country which differs significantly from the US in many respects (e.g., politics, economy, culture) may shed additional light on this link. The lack of evidence on the role of intra-couple financial management in non-western countries with lower levels of financial inclusion has been recently emphasized by van Raaij, Antonides, & de Groot [20].

## Theoretical framework

This study is guided by the framework of Couples and Finances Theory (CFT) [19]. At the core of the CFT is the prediction that difficulties encountered in intra-household financial management are related to couple relationship problems and thus the perception of quality of life. The theoretical framework of CFT draws on the systems theory and, as a result, it views intra-household financial matters and relationship quality as two interdependent subsystems

of a higher-order system. In other words, one important assumption of the theory is that financial difficulties and deterioration in relationship quality may influence each other in a self-powered feedback loop.

From the perspective of the systems theory, both partners in a relationship interact within a system in which the actions of one partner convey information that becomes a basis for making decisions about behavior by the other [21]. The coherence or divergence of approaches to life decisions translates into the lives of individual partners. "In a distilled sense, this interactional spiral, composed of behaviors between the spouses and each one's perception and reception of the information carried by those behaviors, captures the notion of a marital interaction system", as emphasized by White, Martin, & Adamsons ([21], p. 152). In this study, the system approach helps to explain the interdependence of dyadic interaction and financial behavior, two domains or subsystems which function within a larger system, and to examine the context in which they operate.

**Background review.** Literature provides a lot of research on the explanation of factors that are important for well-being, including happiness, life satisfaction and positive affect. Subjective well-being is defined here as an emotional and cognitive assessment of life. It includes emotional reactions to given events, but also cognitive assessments and judgments regarding fulfillment and satisfaction [22]. It is a broad concept that includes experiencing pleasant emotions, low levels of negative moods and high levels of life satisfaction [22]. The quality of a relationship is important factor for the quality of life. The concept of relationship quality and well-being is interdisciplinary in nature and thus many theoretical constructs have been developed. Among the variables that characterize the relationship quality, previous studies focused on marital happiness [9,23], adjustment (cf. [24,25]), success [2,26], satisfaction (cf. [27–30]), and quality (cf. [31,32]). In this study, relationship quality is assumed to be a unifying term and multi-dimensional phenomenon which includes happiness, adaptation and communication, satisfaction, and a sense of integration [32]. Therefore, the literature review section also presents research based on related concepts.

Literature on financial planning and counselling, and on the psychology and sociology of marriage and family shows a link between relationship satisfaction and financial management in dyadic settings (also in terms of disagreements). In the review of effective factors affecting relationship quality, two groups of variables were particularly taken into account: (i) those associated with interactional dynamics in a couple (also in the context of financial life of a family), (ii) those related to financial management. In this study we primarily focus on the interactional factors, understood here as shared goals as well as gender differences within the relationship.

According to Archuleta [19], similar financial goals, values regarding the importance of money in life, autonomy, independence, and aspirations affect marital satisfaction. In addition, when partners share similar views on their roles in the relationship, they are satisfied with their individual management roles of mother, father, wife, husband, etc. They also perceive their finances as properly managed. The study in question shows further that marital satisfaction is higher when the relationship involves shared meanings and respect for the partner's life dreams and plans [19].

Among the factors which adversely affect the quality of life, financial stress is responsible for consistent negative effects in relationships [33]. Financial problems have an impact on a couple's ability to communicate and resolve conflicts they have [17,34]. Couples who believed in marriage for life and who shared decisions equally with their partners reported low conflict more often and had a high level of well-being [35].

Previous research also shows that both satisfaction with marriage and its stability stem from dyadic processes: the partners perceive their ability to connect and interact, which has an

impact on relationship satisfaction [36]. Some of these results suggest that relationship dynamics differ between the female and the male member of a couple.

It can be therefore stated that interaction strategies of individuals play a role in the perception of relationship quality. Hence, we hypothesized that having common goals and shared values should be directly related to this perception. When members of a couple have shared beliefs about finances, they experience higher satisfaction with the relationship, and hence arguing about daily money problems is likely to be reported less frequently. Due to the fact that these factors are interactional in nature, it is important to compare declarations from both members of a couple.

Financial management behaviors (and their perception) as well as financial problems (and their perception) have also a significant relationship with marital satisfaction [37]. According to a qualitative study conducted by Skogrand, Johnson, Horrocks and DeFrain [38], most of the couples who described themselves as "great marriages" practiced the model where one of the partners was in charge of everyday finances. This division of duties required trust and proper communication. Moreover, these couples had little or no debt, or their goal was to pay off debt. They did not go beyond their means and were thrifty. The study indirectly shows that satisfaction with financial decisions in general is more important than whether couples make them together (and agree on them) [38]. A vast amount of empirical data from studies conducted in Poland and presented by Plopa [39,40] indicates that personal competencies of relationship partners are essential, and that their subjective resources are significantly related to their level of satisfaction with their relationship.

In their study investigating financial management practices of student couples, Rea, Zuiker, & Mendenhall [41] conclude that relationship communication is an important factor during life changes. Their key findings suggest that young people believe that communicating about finances helps to prevent or solve financial challenges [41]. Apart from research showing the importance of healthy financial management for relationship satisfaction, there are studies confirming that financial issues are a significant cause of disagreements between partners [42,43]. It should be noted that, as observed, relationship partners did not consider money the most common source of conflict between them. However, considering other, non-financial reasons, conflicts about money were more pervasive, problematic, and repetitive [43].

Britt et al. [44] also point to the link between financial issues and the quality of relationships. They present how relationship satisfaction and perceived spending behaviors are interrelated. According to their findings, spending behaviors on the part of the partner affect relationship satisfaction. Interestingly, such dependency does not apply to one's own spending behaviors or those performed jointly. This indicates the need to analyze reports from both members of a couple and see how the financial behaviors of one partner are related to the perceived satisfaction of the other.

Research has shown that relationship quality is affected by stress over economic matters. Economic hardship is most strongly related to thoughts about divorce [36]. Financial behaviors performed by each partner and those performed jointly are one of the main reasons for relationship dissatisfaction, which may lead to the break-up of relationship or to divorce (cf. [44]). Therefore, we hypothesized that financial management behaviors impact on relationship quality.

One of the conclusions of the study conducted by Conger et al. [45] indicates that hostility between marriage partners caused by economic pressure reduces the quality of their marriage. In addition, conflict management skills mediated the relationship between the impact of economic pressure and the quality of relationship.

Research shows that when it comes to satisfaction with marriage, it also matters who manages the money. Managing financial resources can be a sign of power in a household. Some

household money management systems reflect traditional perceptions of social roles of men and women. Interestingly, it has been established that women's level of satisfaction with their financial situation increases when the adopted system is characterized by individual, separate responsibility for expenses. Some researchers emphasize that new individualized models of money management actually contribute to better living standards and increased financial well-being of women [46].

In contrast, men's financial control increases their financial satisfaction (regardless of women's income). Gender norms in financial management indicate that dealing with financial matters can have different meanings for women and men. Men's financial control is associated with comfort or security by re-establishing a traditional balance [46]. The results of a study conducted by Yucel [47] indicate the need to construct two models, separately for women, separately for men. The author indicates that combining both partners' reports of relationship satisfaction offers a worse fit to the data. Therefore, we hypothesized that financial management behaviors of one relationship partner would be important for the relationship quality as perceived by the other. Moreover, cohesion or divergence in the field of financial management between partners should be taken into account.

## Materials and methods

### Sample and data

The empirical material for this study has been collected in December 2018 from 500 heterosexual couples (1,000 adult respondents) using an online questionnaire (computer-assisted web interviewing). Ethical approval was granted by the Bioethical Commission at the WSEI University in Lublin. The data were analyzed anonymously. The data collection phase was handled by DRB Research, a professional market and opinion research agency. The sample was controlled by cross-section quotas based on the European Union NUTS2 administrative division. All respondents answered a series of questions related to different aspects of quality of life (The Well-Matched Marriage Questionnaire (KDM-2), Shared Goals and Values and Harsh Start-Up) and financial management behaviors (Cash Management, Savings and Investment, Credit Management, Insurance). Couples with the older male partner dominated (almost 63%) over those with the older female partner (nearly 13%). More than three-quarters of couples were formally married (77%), while less than one-quarter were cohabiting (23%). More descriptive statistics can be found in Table 1.

Due to the specific population of respondents we were not able to establish a sampling frame from which a random sample could be drawn. Consequently, our sample is purposive not random. Although the purposive sampling does not yield the representativeness of results that could be projected to the population of dating couples in Poland, there are research showing that the non-probabilistic methods have similar levels of accuracy as probabilistic ones [48].

Due to the rigorous data collection procedure (dyadic matched-pairs), we found that in some cases observations for some variables used in this study were missing. Although those observations were missing at random, the number of complete cases–couple as a unit–decreased considerably (8% of missing values in total; eight variables). Instead of wiping out potentially useful datapoints (listwise deletion), we tried to reconstruct plausible values for missing responses reflecting the relations between members of couples. We applied three imputation methods (adjusted to our variables) offered by R 'mice' package [44]: (i) random forest (rf), (ii) weighted predictive mean matching (midastouch) (iii) proportional odds model (polr). The weighted predictive mean matching (midastouch) returned the lowest coefficient of variations, therefore later in this study we build on imputation results of this procedure. The weighted predictive mean matching (midastouch) returned the lowest averaged (for eight

**Table 1. Descriptive statistics of the sample.**

| | Women * | Men * |
|---|---|---|
| Age (avg) | 41.3 | 43.6 |
| Years of partnership (avg) | 23.2 | 23.4 |
| Dependent children (avg) | 1.09 | 1.096 |
| Financial dependency on parents (%) | 15 | 14 |
| Level of education | **N** | **N** |
| Primary | 4 | 8 |
| Lower secondary | – | 3 |
| Basic vocational | 32 | 61 |
| Secondary | 68 | 40 |
| Secondary vocational | 79 | 123 |
| Post-secondary | 67 | 47 |
| Higher | 236 | 198 |
| At least PhD | 14 | 20 |
| Monthly salary (in PLN) | **N** | **N** |
| < 1,500 | 85 | 29 |
| 1,500–2,500 | 148 | 91 |
| 2,501–3,500 | 151 | 152 |
| 3,501–4,500 | 64 | 109 |
| 4,501–6,000 | 30 | 67 |
| > 6,000 | 22 | 52 |
| Marital status | **N** | **N** |
| Married (first) | 355 | 354 |
| Married (another) | 29 | 29 |
| Cohabitation (first) | 78 | 75 |
| Cohabitation (another) | 38 | 42 |

* Women and men subpopulations sum up to 1000. All categorical variables sum up to 500 in both subpopulations.

variables with missing data) root mean square error (RMSE), therefore later in this study we build on imputation results of this procedure [49].

## Measures

**Shared goals and values.** The Shared Goals and Values Scale [18] is an adapted four-item measurement drawing on Gottman's [50] Shared Meaning Roles, Shared Meaning Goals, and Shared Meaning Symbols scales, which are used to assess couples' shared meaning of financial goals and values, life goals, and autonomy. The items were measured using a 7-point Likert-type scale, where 1 = strongly disagree and 7 = strongly agree. Response scores could range from 4 to 28, with lower scores indicating lower agreement on life goals and values, and higher scores reflecting more agreement on these issues. The four statements included in this scale were as follows: (a) "We have similar financial goals"; (b) "Our hopes and aspirations, as individuals and together, for our children, for our life in general, and for our old age are quite compatible"; (c) "We have similar values about the importance and meaning of money in our lives"; (d) "We have similar values about 'autonomy' and 'independence'.

*Harsh start-up.* Harsh start-up was measured using a scale consisting of five items. They were adapted from a work originally published by Gottman & Silver [51] and translated into Polish. Conceptually, harsh start-up can be viewed as a way in which couples interact; more

specifically, it reflects how couples engage in the discussion process concerning conflictual topics. Each of the following items was assessed dichotomously, with a true statement assigned the score of 1, otherwise 0. The items were reversely coded and summed into a harsh start-up index scale score so that higher scores reflected being less likely to engage in harsh start-up. The items were as follows: (a) "Arguments often seem to come out of nowhere", (b) "I seem to always get blamed for issues", (c) "My partner criticizes my personality", (d) "Our calm is suddenly shattered", (e) "I think my partner can be totally irrational".

*Well-Being.* SWLS was measured using a scale consisting of five items statements. Respondents assess to what extent each of them relates to their lives. The result of the measurement is a general indicator of a well-being understood as sense of life satisfaction, specifically global cognitive judgments of satisfaction with one's life [52]. The SWLS asked the respondents to rate on a 5-point Likert-type scale (1—I strongly disagree, 5 –I strongly agree), the extent to which they agree with statements, e.g. "In most ways my life is close to my ideal", "The conditions of my life are excellent", "If I could relive my life", "I would change almost nothing". The Polish translation of SWLS has been used and shown to have strong internal reliability.

**Relationship quality.** The Well-Matched Marriage (KDM-2) questionnaire [40,53] was used to measure the quality of the relationship from the perspective of four dimensions: intimacy, disappointment, self-realization, and similarity as well as the overall result indicating overall satisfaction with the marriage/relationship. The tool has satisfactory psychometric indicators concerning research on the population of Polish marriages and couples. Cronbach's alpha for individual subscales ranges from 0.81–0.89. It is the only psychometrically validated scale that has been validated in a nationally representative sample scale that has been designed in Poland to date.

The KDM-2 questionnaire applies to both partners individually and to couples. In this study, the questionnaire was adopted to also examine cohabitation relationships and consists of 32 statements. The respondent, while answering the questions, is asked to choose one of five answers on a scale from 1 = totally disagree to 5 = totally agree (5-point Likert-type scale).

**Financial management behavior.** Following Shim et al. ([54], p. 1459), in this article we conceptualize healthy financial behavior as "the broad set of desirable behaviors that help (. . .) achieve the financial, economic, and inter-personal goals". To operationalize this concept, we used the Financial Management Behavior Scale (FMBS) developed by Dew and Xiao [29]. The scale, validated psychometrically in a sample representative for the adult US population by [29], has been recently adopted by other researchers in its original or modified form [55–59].

The scale is a multidimensional instrument consisting of four subscales: (a) Cash Management, (b) Savings and Investment, (c) Credit Management, and (d) Insurance. We used three of these subscales in their original form, except for the Credit Management subscale. We decided to construct this subscale from scratch, because Dew and Xiao's [29] instrument is leaned towards credit card behavior, which is warranted given the specificity of the US credit market, where credit cards are very common. However, the strong reliance of the subscale on credit cards hardly reflects the borrowing practices of consumers in Poland. As reported by the Polish Bank Association [60], only one in six adult Poles has a credit card.

Development of the revised credit management subscale has been demonstrated in detail in A. Cwynar, W. Cwynar, M. Baryła-Matejczuk, & M. Betancort [61]. The following items have been included in the scale: (i) "I compared offers before applying for credit", (ii) "I got behind on debt repayment, including interest on debt", (iii) "I borrowed to repay existing debt", (iv) "I borrowed simultaneously from more than one source (e.g., banks, personal loan/payday loan companies, instalment purchases, pawnshops, family etc.)", (v) "I borrowed for at least one of the following purposes (or for similar purposes): the purchase of expensive clothing or accessories (e.g., a branded suit or purse), a holiday abroad, technological novelties or gadgets", (vi) "I

made only minimum payment on a loan" and (vii) "I paid off credit card balance in full each month".

To obtain data on financial management behavior, the respondents were asked to answer the following question: "On a scale from 1 = never to 5 = always, indicate how often you have engaged in the following activities in the past six months" (the exact wording of all items comprising the FMBS can be found in Dew and Xiao's [62] original article, except the credit management subscale, whose constituent items have been listed above. Given that some behaviors included in the credit management subscale are unambiguously undesirable and should be avoided, the respondents' reports regarding these behaviors were reversely coded. As a result, the outcomes measured on each subscale and the overall score on the FMBS can be interpreted in a straightforward way: the higher the value on the scale, the more sound the financial behavior.

## Analyses

We built the actor-partner interdependence models (APIM) that offer in-depth insights into the dyadic relationships between female and male partners [63]. Instead of analyzing two individuals as nonindependent actors, we treat them as unit being nested within the dyad. To meet this goal we specified a path model in which life satisfaction reported by male and female partner were predicted simultaneously by each actor's and hers/his partner's: KDM, hared Goals and Values, Harsh Start-up; four financial management behavior variables: Cash Management, Savings and Investment, Credit Management, and Insurance.

Important question regarding this type of models is whether actors within a dyad may be considered indistinguishable, in our case with respect to gender. That means, whether determinants of well-being are the same, and their effects are of the same size, for both men and women. To test this assumption, we estimated two variants of our model: in one we imposed constraints on model parameters to hold values of analogous regression coefficients in the equation for men and the equation for women the same. In other one we enabled parameters to vary between actors of different gender. The result of the chi-squared test ($chi^2(12) = 26,789$, $p = 0.05$) enables us to reject the hypothesis of actors' indistinguishability [57,58]. Therefore, the model proposing the distinguishability with respect to gender, i.e. that there are differences in determinants of well-being between men and women, has been further discussed.

Due to large number of independent variables we decided to test only actor and partner effects in our model. Interaction effects often analyzed in APIM models have been deliberately omitted to simplify an already complex model.

## Results

The series of the actor effects were tested for the purpose of this research. Four significant relations has been observed in this dimension. The results of the analysis are presented in Table 2.

The shared goals and values declared by both woman and man has a positive impact on her and his well-being. Moreover, woman's credit management has a positive impact on her well-being. There is also a week negative relationship between the quality of the relationship declared by man and dependent variable. Surprisingly, higher level of the quality of the relationship, the lower level of his well-being.

For partner effects, women's 'Shared Goals and Values' variable has a significant but negative impact on men's assessment of well-being. At the same time, women's save and investment behavior positively affects men's well-being. It is also worth noting that men's "Insurance" variable is not far from the commonly accepted statistical significance threshold and may negatively affect women's well-being.

**Table 2. APIM model effects.**

| APIM paramters | | Estimate | Z | P-value |
|---|---|---|---|---|
| **Actor effects** | | | | |
| 1 | Well_B_M ~ KDM_M | -0.020 | -1.797 | 0.072. |
| 2 | Well_B_M ~ SGV_M | 0.373 | 5.659 | 0.000*** |
| 3 | Well_B_M ~ Harsh_M | -0.207 | -1.209 | 0.227 |
| 4 | Well_B_M ~ Save_Inv_M | -0.057 | -0.969 | 0.332 |
| 5 | Well_B_M ~ Insurance_M | 0.059 | 0.557 | 0.577 |
| 6 | Well_B_M ~ Credit_Man_M | -0.003 | -0.027 | 0.978 |
| 7 | Well_B_M ~ Cash_Man_M | -0.138 | -1.418 | 0.156 |
| 8 | Well_B_W ~ KDM_W | 0.007 | 0.628 | 0.530 |
| 9 | Well_B_W ~ SGV_W | 0.224 | 4.008 | 0.000*** |
| 10 | Well_B_W ~ Harsh_W | -0.388 | -2.286 | 0.022 |
| 11 | Well_B_W ~ Save_Inv_W | 0.067 | 1.427 | 0.154 |
| 12 | Well_B_W ~ Insurance_W | 0.151 | 1.508 | 0.132 |
| 13 | Well_B_W ~ Credit_Man_W | 0.216 | 2.231 | 0.026* |
| 14 | Well_B_W ~ Cash_Man_W | -0.068 | -0.880 | 0.379 |
| **Partner effects** | | | | |
| 15 | Well_B_M ~ KDM_W | 0.017 | 1.521 | 0.128 |
| 16 | Well_B_M ~ SGV_W | -0.199 | -3.325 | 0.001** |
| 17 | Well_B_M ~ Harsh_W | -0.086 | -0.474 | 0.635 |
| 18 | Well_B_M ~ Save_Inv_W | 0.171 | 3.381 | 0.001** |
| 19 | Well_B_M ~ Insurance_W | -0.012 | -0.113 | 0.910 |
| 20 | Well_B_M ~ Credit_Man_W | 0.015 | 0.144 | 0.886 |
| 21 | Well_B_M ~ Cash_Man_W | 0.110 | 1.329 | 0.184 |
| 22 | Well_B_W ~ KDM_M | 0.004 | 0.385 | 0.700 |
| 23 | Well_B_W ~ SGV_M | 0.065 | 1.053 | 0.292 |
| 24 | Well_B_W ~ Harsh_M | 0.004 | 0.027 | 0.978 |
| 25 | Well_B_W ~ Save_Inv_M | 0.087 | 1.586 | 0.113 |
| 26 | Well_B_W ~ Insurance_M | -0.182 | -1.824 | 0.068. |
| 27 | Well_B_W ~ Credit_Man_M | -0.138 | -1.396 | 0.163 |
| 28 | Well_B_W ~ Cash_Man_M | -0.008 | -0.085 | 0.933 |

Note: Significance Codes

*** < 0.001

** < 0.01

* < 0.05; 0.05–0.099.

## Discussion

This study forms a part of the research devoted to exploring the connection between the individual well-being and a certain set of variables describing the financial management behavior and interaction in couples. We made an attempt to broaden the existing range of knowledge concerning the way in which the financial behavior, the quality of the relationship, and the sense of common goals and values affect the well-being as reported both by her or him as well as by the other partner. Several actor and partner class effects, characteristic to the APIM analytical approach [63], were identified. The obtained results suggest that the well-being declared by a man may be positively associated with the save and investment, as well as, interaction factors of a woman. Similarly, male insurance behavior has negative impact on women's well-being. However, this impact is not statistically strong, as it goes beyond the traditionally

understood significance level. Nevertheless, such results may suggest that the financial behavior of one partner can affect the well-being of the other side of the relationship.

Among the partner effects, the negative impact of shared goals and values (SGV) of woman on men's well-being is noticeable. In other words, the higher agreement on life goals and values declared by woman, the lower well-being reported by men. By contrast, the variable SGV reported by men does not affect the well-being indicated by women. Such result may suggest that life goals and values indicated by women and men do not always coincide in the sample we analyzed but only men declared some discomfort with SGV stated by women. Furthermore, it is worth noting that other interactional variables e.g. harsh start-up or relationship quality declared by one side of the relationship had no effect on the other side. If we pay attention to the nature of this influence, we may observe that the woman's belief in common values and goals at the financial level has a negative impact on the satisfaction with life as reported by the partner. If a woman believes that she and her partner share similar financial goals, and that they agree on their hopes and aspirations for living together, and have a similar approach to the values placed on the importance of money in their lives, the sense of satisfaction with life is lower from the man's point of view. The autonomy and independence of a woman in this respect carries a negative perception of the partner according to his perception of the subjective well-being. This is in line with the system approach, according to which the behavior of one partner is important for the quality of life as experienced by the other [21].

In the model estimated we also found that the men's score at the SGV scale is positively associated with his well-being (actor effect). Such a result is rather complementary to the relationship presented above. At the same time, similar relationship between SGV and well-being has been identified for women. Such a relationship may suggest that the convergence of life goals and values is a significant component of both women and men's well-being.

Surprising, the quality of the relationship indicating overall satisfaction with the marriage declared by men has negative impact on his well-being. This result has a specific, locally-determined character that relates to Polish or Eastern European conditions.

As to the role of variables describing financial management behavior (financial variables, henceforth), insurance and credit management behavior come to the fore. The credit management of women is associated with their well-being. When it comes to mutual interaction, women's saving and investing behavior has a positive relationship with men's declared well-being. On the other hand, men's insurance-related behavior has a negative relationship with women's well-being. In the discussion presented in this section we have assumed that the extent to which the financial behavior is sound (desirable, healthy) depends on the partner's involvement in dealing with household financial matters. Such a view is consistent with the concept of learning by doing [64] and experiential learning [65] theories and has been confirmed empirically [20,66,67]. Simply put, individuals who are more involved in managing intra-household finances are more likely to engage in sound financial behavior due to feedback stemming from experience. Our results, seen from this perspective, may suggest that couples in Poland appreciate a division of financial labour (that is, a specialization in the domain of household financial management)–at least in some dimensions of their management.

In the results obtained, special attention should be paid to the roles of self-description—statements regarding common goals have the character of declarations. The need for security and independence associated with saving is emphasized, especially if practiced by women. Moreover, if we look at the coexistence of variables, we can draw a very cautious conclusion that a higher quality of life for women is associated with a lower need for insurance by men. The motives for making decisions about insurance can be more complex than it seems [68]. It may also be that the sense of comfort and security of women means that they will not put pressure on a man who is stereotypically responsible for finance, especially

in Poland (cf. [69]). Another area we pay attention to is the experience of women in financial management (cf. [70]).

The obtained result may indicate the specificity of Polish society at this stage of its socio-economic development. The less experience people have in resource management and with ideas about women living more independent lives, the more such factors can pose a threat to the quality of men's relationships. On a cautionary note, such a hypothesis requires confirmation through further research. Seeking security in the form of insurance may be a manifestation of insecure behavior rather than a process of securing a degree of safety.

Our research also does not support the previous studies [69,70] that women more often take responsibility for short-term financial management (shopping, paying bills, managing the current budget–all of which are part of cash management in the financial management behavior scale that we applied in our study), while men are more often responsible for long-term financial decisions (that is, those related to savings and investment, credit management and insurance in the financial management behavior scale).

## Limitations and future research

There are certain limitations to this study that could be improved in future research. First, our sample was not random due to the complex sampling procedure. Thus, the results cannot be projected to the entire population of married and cohabiting couples in Poland but rather indicate interesting associations between cross-sectional variables used in our study. It is therefore probable that future research based on random samples will bring more in-depth and comprehensive results.

Second, the selection of variables related to different aspects of financial issues was primarily based on our research experience as there is no similar cross-sectional (psycho-economic) research in the literature. Although some of our results seem promising we believe that international comparative studies would contribute to the state of knowledge about the relationship between various psychosocial and economic variables.

Third, due to certain cultural conditions shaping the quality of relationship in married and cohabiting couples in Central Eastern Europe, our results may have unique but rather local character. The replication of research itself can be very difficult, even though the issues we were focused on are universal and timely in nature. It seems therefore that conducting similar research in different cultural settings could increase our knowledge about the well-being of individuals and couples dealing with various financial decisions and problems.

In addition, in the planned future research, declarative areas need to be clarified, translated into specific situations and even an experimental model. This means that perhaps the declaration regarding living together with a partner has its limitations, which do not allow to verify whether there is communication in the relationship or whether it is only a subjective perspective of one of the partners. Including the relationship communication variable would be important and would help with a more objective rather than declarative indicator of relationship life.

## Author Contributions

**Conceptualization:** Monika Baryła-Matejczuk, Wiesław Poleszak, Andrzej Cwynar.

**Data curation:** Kamil Filipek.

**Formal analysis:** Kamil Filipek, Tomasz Żółtak.

**Investigation:** Monika Baryła-Matejczuk, Wiesław Poleszak, Andrzej Cwynar.

**Methodology:** Monika Baryła-Matejczuk, Andrzej Cwynar.

**Software:** Kamil Filipek, Tomasz Żółtak.

**Supervision:** Monika Baryła-Matejczuk.

**Validation:** Monika Baryła-Matejczuk, Wiesław Poleszak.

**Writing – original draft:** Monika Baryła-Matejczuk, Wiesław Poleszak, Kamil Filipek, Andrzej Cwynar.

**Writing – review & editing:** Monika Baryła-Matejczuk, Andrzej Cwynar, Tomasz Żółtak.

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
