## [Decision Letter · Decision Letter 0]

29 Apr 2022

PONE-D-21-25538In the quest for effective factors of relationship quality: Insights from intra-couple interaction and financial management variablesPLOS ONE

Dear Dr. Baryła-Matejczuk,

Thank you for submitting your manuscript to PLOS ONE. After careful consideration, we feel that it has merit but does not fully meet PLOS ONE’s publication criteria as it currently stands. Therefore, we invite you to submit a revised version of the manuscript that addresses the points raised during the review process.

The expert reviewer raises a number of concerns regarding the methodological reporting of the manuscript. In particular, they request further clarification and justification for the procedures and analyses reported.

Can you please address these concerns in your revision?

We look forward to receiving your revised manuscript.

Kind regards,

Avanti Dey, PhD

Staff Editor

PLOS ONE

Reviewers' comments:

Reviewer's Responses to Questions

**Comments to the Author**

1. Is the manuscript technically sound, and do the data support the conclusions?

Reviewer #1: Partly

2. Has the statistical analysis been performed appropriately and rigorously? 

Reviewer #1: No

3. Have the authors made all data underlying the findings in their manuscript fully available?

Reviewer #1: Yes

4. Is the manuscript presented in an intelligible fashion and written in standard English?

Reviewer #1: No

5. Review Comments to the Author

Reviewer #1: This paper describes a study investigating the associations among financial management behaviors and relationship quality in heterosexual couples in Poland. The introduction needs to be "smoothed" out with less repetition. I suggest beginning with the theory, then provide a literature that does or does not report the theory and then what this study will show about the theory.

My major concerns are with the analyses:

1) A smaller concern is that I could not figure out what the instrument was for relationship quality, that needs to be made clearer, what were the items?

2) Although it was appropriate to separate the males and females given the nonindependence of the data and adding the other partner's satisfaction was important, I think a dyadic data analysis process would have been more appropriate. I was also missing a correlation table showing the correlations of the variables both within and between partners. I worry that the "flip" in the direction of the relationships is more about multicolinearity than reality. Cook and Kenny have shown these "flips" when including multiple family members' data in the same regression equation.

3) I don't understand why an Actor Partner Interdependence model was not used. In the APIM both partner's data and both partner's outcomes are included in the model and actor and partner paths can be tested for equivalence. Using step wise methods are problematic for testing theory, and I don't think it is appropriate to discuss gender differences without also testing the estimates for equivalence. I think the APIM is a much more appropriate model to test and would provide a better test of the theory, than what was presented here.

4) Given the fact that we don't know if the differences between the equation for males and equation for females were statistically significant, and issues with multicollinearity, I don't know that I would trust the discussion section. I would rather see an APIM model with the actor and partner effects tested for equivalence between the partners, before I would be able to trust that the gender differences shown were actual differences and not artifacts of the analyses.

6. PLOS authors have the option to publish the peer review history of their article (what does this mean?). If published, this will include your full peer review and any attached files.

Reviewer #1: **Yes: **Suzanne Bartle-Haring

---

## [Author Response · Author response to Decision Letter 0]

21 Jul 2022

Response to Reviewer #1 Comments

Point 1: This paper describes a study investigating the associations among financial management behaviours and relationship quality in heterosexual couples in Poland. The introduction needs to be "smoothed" out with less repetition. I suggest beginning with the theory, then provide a literature that does or does not report the theory and then what this study will show about the theory. 

Response 1: Thank you for your suggestions to improve this manuscript. The theoretical part has been rebuilt, the introductions have been shortened in order to keep the essence, the theoretical basis begins the article, successively reports from the literature and research were recalled explaining the direction taken in the prepared analyses.

Point 2: My major concerns are with the analyses:

2.1 A smaller concern is that I could not figure out what the instrument was for relationship quality, that needs to be made clearer, what were the items?

Response 2.1: Thank you for your comment, the Well-Matched Marriage (KDM-2) questionnaire was used to measure the quality of the relationship (source: Kaźmierczak M, Plopa M. Communication in Marriage Questionnaire – Conclusions from Marital Studies. Pol Forum Psychol. 2006; 11(2): 213-226.). The questionnaire has a four dimensions: intimacy, disappointment, self-realization, and similarity as well as the overall result indicating overall quality of relationship. In our analysis, we took into account the overall result. Questions that go into the tool include i.a.:

Marriage / relationship life with its struggles is only the authentic life that every human being should experience.

In our free time, we try to be together.

As the years go by, the contact between us deepens and we feel more and more connected with each other.

I regret the lost independence, freedom from the pre-marriage period.

When we have a sharp exchange of views, this fact causes a long-lasting disconnect.

We agree when it comes to spending free time, weekend and holidays.

As the marriage / relationship lasts, we are more sensitive to our own needs, more subtle, mature in mutual contacts.

Relationship / marriage has become an obstacle for me in achieving my own goals, for example professional aspirations.

Joint activities and the implementation of joint plans bring me satisfaction.

I feel better at work than at home.

I can say that I found happiness in marriage / happiness in a relationship.

I feel lonely in a marriage / relationship.

Our worldview coincides.

As the examples above shows, the questionnaire consists of statements about marriage / relationship, some of them refer directly to a person as a partner in a marriage / relationship, while others refer to marriage / relationship as a whole.

2.2 Although it was appropriate to separate the males and females given the nonindependence of the data and adding the other partner's satisfaction was important, I think a dyadic data analysis process would have been more appropriate. I was also missing a correlation table showing the correlations of the variables both within and between partners. I worry that the "flip" in the direction of the relationships is more about multicolinearity than reality. Cook and Kenny have shown these "flips" when including multiple family members' data in the same regression equation.

2.3 I don't understand why an Actor Partner Interdependence model was not used. In the APIM both partner's data and both partner's outcomes are included in the model and actor and partner paths can be tested for equivalence. Using step wise methods are problematic for testing theory, and I don't think it is appropriate to discuss gender differences without also testing the estimates for equivalence. I think the APIM is a much more appropriate model to test and would provide a better test of the theory, than what was presented here.

Response 2.2&2.3: Thank you for the suggestion of improvement and the opportunity to refer readers to other studies. We have built two APIM models as presented by Cook and Kenny. The result of the chi-squared test (chi^2(12) = 24.039, p = 0.02) enables us to reject the hypothesis of actors indistinguishability. Therefore, the model proposing the distinguishability with respect to gender has been further discussed in the manuscript. Thanks to this great suggestion, we were able to identify both actor and partner effects. As a consequence sections: Analysis, Results and Discussion have been modified accordingly. 

 Unfortunately, no one at our research team had experience with APIM models. We therefore decided to invite an experienced researcher to help us prepare APIM models. After more than a dozen attempts, we were able to estimate two models that shed new light on the relationships analysed in the text. However, there was a change in the independent variables. The harsh start-up variable spoiled all the models, so we decided to replace it with a well-being variable, which (according to the theoretical approach) very often appears in analyses of different couple relationships. All the changes made have been marked accordingly in the text. 

2.4 Given the fact that we don't know if the differences between the equation for males and equation for females were statistically significant, and issues with multicollinearity, I don't know that I would trust the discussion section. I would rather see an APIM model with the actor and partner effects tested for equivalence between the partners, before I would be able to trust that the gender differences shown were actual differences and not artifacts of the analyses. 

Response 2.4 The discussion section has been reworked based on data from the APIM model. The results shed new light on the analysed relations and constitute material worth extending with further research. The results obtained with the APIM model were presented in the Results section, and their interpretation appeared in the Discussion section.

---

## [Decision Letter · Decision Letter 1]

19 Sep 2022

PONE-D-21-25538R1In the quest for effective factors of relationship quality: Insights from intra-couple interaction and financial management variablesPLOS ONE

Dear Dr. Baryła-Matejczuk,

Thank you for submitting your manuscript to PLOS ONE. After careful consideration, we feel that it has merit but does not fully meet PLOS ONE’s publication criteria as it currently stands. Therefore, we invite you to submit a revised version of the manuscript that addresses the points raised during the review process.

We look forward to receiving your revised manuscript.

Kind regards,

José Alberto Molina

Academic Editor

PLOS ONE

Reviewers' comments:

Reviewer's Responses to Questions

**Comments to the Author**

1. If the authors have adequately addressed your comments raised in a previous round of review and you feel that this manuscript is now acceptable for publication, you may indicate that here to bypass the “Comments to the Author” section, enter your conflict of interest statement in the “Confidential to Editor” section, and submit your "Accept" recommendation.

Reviewer #1: All comments have been addressed

Reviewer #2: (No Response)

Reviewer #3: (No Response)

2. Is the manuscript technically sound, and do the data support the conclusions?

Reviewer #1: Yes

Reviewer #2: Yes

Reviewer #3: Partly

3. Has the statistical analysis been performed appropriately and rigorously? 

Reviewer #1: Yes

Reviewer #2: Yes

Reviewer #3: Yes

4. Have the authors made all data underlying the findings in their manuscript fully available?

Reviewer #1: Yes

Reviewer #2: Yes

Reviewer #3: Yes

5. Is the manuscript presented in an intelligible fashion and written in standard English?

Reviewer #1: Yes

Reviewer #2: Yes

Reviewer #3: Yes

6. Review Comments to the Author

Reviewer #1: The authors have done a great job responding to the original review. I appreciate their finding an expert in dyadic data analysis and testing for indistinguishability. There are also tests that can be performed on the paths to determine if they are statistically different, but I can let that go in this version.

Reviewer #2: The investigators have built two APIM models as presented by Cook and Kenny with rejection of the hypothesis of actors indistinguishability. Assuming the accuracy of the fit as the investigators note, the results are very brief and descriptive and appear to follow from the modeling.

There are certain limitations to this study which the authors note. Primarily, the sample was not random due to the complex sampling procedure. Thus, the results cannot be projected to the entire population of married and cohabiting couples in Poland but rather indicate possible interesting associations only between cross-sectional variables used in this study

Reviewer #3: Point 1 – Research question and scope

"Do financial management behaviors of one partner have an impact on the perception of relationship quality by the other?" I'm not sure this research question can be answered by the current design since causality cannot be inferred. For example, the current research would yield the same results with reverse causality - if one feels that their spouse perceives their relationship to be low quality (i.e., they might break up), then they might alter their financial management behavior (e.g., because the future is uncertain). Therefore, what would happen to the relationship quality if a partner improved their financial management is unclear. In fact, the authors are aware of that when they describe the "feedback loop" in line 85. However, the lack of causality here also allows for other factors to be at play. For instance, if one spouse feels unwell, that might affect both partners' financial behavior and perception of relationship quality, which would confound the analysis. Thus, the authors address the following question "Do financial management behaviors of one partner correlate with the perception of relationship quality by the other?" One way to achieve causality could be to randomize feedback for financial management – i.e., participants describe their own and partners' financial behavior, then they receive feedback that is either positive or negative (randomly), then they report relationship quality.

Point 2 – Data access

(https://data.mendeley.com/datasets/wbbgngvdvr, the DOI link is broken for me)

Since the study is published in English, I think it will be helpful for other researchers to have the data translated. The polish version should be kept, but variable names and question translations are necessary to understand the data.

Point 3 – Measures of relationship quality

It would be interesting to see measures of relationship quality beyond the perception questionnaire. For example, how do financial management behaviors correlate with the probability of divorce? With the likelihood of having children? Tracing the participants for a follow-up survey could answer these questions and test the correlations over time.

Point 4 – Sample

Five hundred couples answered the survey, but how many were contacted and didn't complete it? Even if it's a representative sample demographically, attrition could create a biased sample, making it useful to report.

Point 5 – Imputation

"The weighted predictive mean matching (midastouch) returned the lowest coefficient of variations." It is not clear to me that imputations should minimize variations. Instead, a good way to test the performance of the imputation models is to use some of the non-missing data as a test group. That is, say there are 100 complete observations (I didn't find the number of observations with missing values?), use 80 to train the model to predict each of the missing variables, then see what it predicts for the 20 observations left out. Finally, evaluate the model's accuracy (AUC/ROC for continuous, precision/recall for discrete) by comparing the test prediction with the actual observed values. 

Point 6 – Well-being questionnaire

It's unclear how many questions were there, what they were about and how they were aggregated.

Point 7 – KDM-2 questionnaire

I didn't understand whether and how the validated questionnaire was modified to examine cohabitation relationships and whether the modifications were validated.

Point 8 – Financial management behavior questionnaire

Measuring how significant financial management is in their lives (beyond how they behave) may be beneficial. For example, it could be that some couples care less than others about finances, and for those couples, we would expect a weaker link between financial management and relationship quality.

Also, that questionnaire seems to measure risk aversion and time preference to a large extent (rather than just financial healthiness and desirability), which may strongly confound the analysis.

Point 9 – Analyses explanation

For a non-APIM expert audience, the analysis is tough to follow. I think there should be some explanation (a paragraph?) of the models' math/statistics and intuition. For example, it's unclear why only these five variables were included in the models (especially the interaction between well-being and shared values). What was the test for which you reported the chi-square comparing exactly (the parameters of the two models?)? What are k and the other model parameters you refer to?

Point 10 – Descriptive analysis

It'd be helpful to see summary statistics about each variable used in the model and their intercorrelations. That will allow understanding of collinearities and interpretation of the main results (e.g., is an effect of 1.8 large?).

Point 11 – Main results

Suppose the model's assumptions and parameters are accurate. The results show that both partners report higher relationship quality when women's cash management questionnaire scores increase. When women's insurance and shared goals and values questionnaire scores increase, their male partner weakly reports lower relationship quality. The effect of the other 20 variables was indistinguishable from zero. Also, I didn't understand how the interactional values arguments (lines 339-340) relate to the discussion, nor the size of the effect and its significance reported.

Also, I think Table 2 should include the number of observations.

It is astounding that no matter how the man is reported to behave financially, it has no relationship with the reported quality of his relationship. I think this and other null effects demand an explanation in the discussion.

 

Point 12 – Conclusions

"Our results, seen from this perspective, may suggest that couples in Poland appreciate a division of financial labour." It might be better to ask that directly in the questionnaire because this does not seem to be a plausible interpretation of the positive parameters for the woman's cash management (most other financial parameters were statistically equal to zero).

"… whereby women deal with current financial affairs, while men are authorized to handle strategic ones." what evidence in the data supports that claim? The only related evidence provided is that when women score higher on the cash management questionnaire (for which I didn't see the questions), both partners report higher marriage quality.

*A few typos I found:

Line 27: "… man's cash management predicts changes in her assessment…"

Line 62: "… pursued hereDo…"

Line 67: "Second, we explore this link using dyadic data, which creates an opportunity to compare reports of both partners.. Fourth,"

7. PLOS authors have the option to publish the peer review history of their article (what does this mean?). If published, this will include your full peer review and any attached files.

Reviewer #1: **Yes: **Suzanne Bartle-Haring

Reviewer #2: No

Reviewer #3: No

---

## [Author Response · Author response to Decision Letter 1]

15 Nov 2022

Dear Editors and Reviewers,

Thank you for the opportunity to revise our paper on ‘In the quest for effective factors of relationship quality: Insights from intra-couple interaction and financial management variables’. Your suggestions have been very helpful for improving the manuscript.

The next round of reviews gave us the impetus to make the essential changes to the text, which we hope have now put our proposal at a very high level. Unfortunately, in the course of working on the next version of the APIM model, we discovered a mistake in the code that put everything we had prepared so far into question. In one line of code, we discovered an unwanted variable, the presence of which affected the quality of the model and, as a result, the value of the ANOVA test. After removing the unwanted variable, the model turned out to be indistinguishable, marking an 180-degree turnaround from the previous (first) version and most importantly, our knowledge of the issue and the theoretical foundations did not allow us to accept this solution. The easiest solution would be to quit and forget, but as experienced researchers we decided to save the project and valuable time of reviewers that has been invested into our manuscript. After discovering a mistake in the code that turned our work upside down we decided to return to the idea that guided us at the beginning of designing the research and writing the text. We changed the dependent variable KDM to well-being, and further enriched the entire model with the variable harsh start-up. That was the original assumption that proved correct at this stage of review.

 We are aware that such things practically do not happen at this stage of the review, but in keeping with the principle of honesty, we decided to inform you about the problems and analytical strategy we have chosen. What could we add?

1. APIM was very new analytical model that we have learnt during the review process. Despite the some great articles describing the use of this approach, translating it into programming code was quite a challenge for us. We decided to invite an additional analyst to cooperate with us, who would be able to prepare the model. The cooperation of two analysts allowed to discover the error in the code.

2. We decided to openly and honestly tell about what happened to us, because we believe that the work of a scientist is often unpredictable, surprising, but also very devastating. You can accuse us of inaccurate preparation of the model after the first wave of reviews, but we emphasize that we get familiar with APIM relatively recently.

3. The model we have prepared now is the result of several dozen hours of work, a choice among several dozen other models, which may prove that it can contribute a lot to widely understood social science.

 Despite the major change we outlined, we are happy that we discovered the error found at this stage. There would be nothing worse than publishing a text that presented the results of an incorrectly built model.

Response to Reviewer #1: 

The authors have done a great job responding to the original review. I appreciate their finding an expert in dyadic data analysis and testing for indistinguishability. There are also tests that can be performed on the paths to determine if they are statistically different, but I can let that go in this version.

Response: Thank you very much for the time and commitment put into the preparation of the first and second reviews for our text. We greatly appreciate the effort and try our best to live up to your expectations.

Response to Reviewer #2: 

The investigators have built two APIM models as presented by Cook and Kenny with rejection of the hypothesis of actors indistinguishability. Assuming the accuracy of the fit as the investigators note, the results are very brief and descriptive and appear to follow from the modeling.

Response: Thank you for all the comments that allowed us to improve the text. It has been a long road to get it to a version that is in line with our research experience and the high standards of a scientific paper. The revised text is the result of our work. Notes on sampling have been added to the text and limitations.

Response to Reviewer #3: 

Point 1 – Research question and scope

"Do financial management behaviors of one partner have an impact on the perception of relationship quality by the other?" I'm not sure this research question can be answered by the current design since causality cannot be inferred. For example, the current research would yield the same results with reverse causality - if one feels that their spouse perceives their relationship to be low quality (i.e., they might break up), then they might alter their financial management behavior (e.g., because the future is uncertain). Therefore, what would happen to the relationship quality if a partner improved their financial management is unclear. In fact, the authors are aware of that when they describe the "feedback loop" in line 85. However, the lack of causality here also allows for other factors to be at play. For instance, if one spouse feels unwell, that might affect both partners' financial behavior and perception of relationship quality, which would confound the analysis. Thus, the authors address the following question "Do financial management behaviors of one partner correlate with the perception of relationship quality by the other?" One way to achieve causality could be to randomize feedback for financial management – i.e., participants describe their own and partners' financial behavior, then they receive feedback that is either positive or negative (randomly), then they report relationship quality.

Response: thank you very much, the text has been corrected according to the note. Attention is drawn to associated than correlated.

Point 2 – Data access

(https://data.mendeley.com/datasets/wbbgngvdvr, the DOI link is broken for me)

Since the study is published in English, I think it will be helpful for other researchers to have the data translated. The polish version should be kept, but variable names and question translations are necessary to understand the data.

Response: thank you for your comments, indeed, a much greater usability of the database is possible thanks to the translation. We have translated the database and upload the current data:

Baryła-Matejczuk, Monika (2022), “Factors of relationship quality/SWLS ”, Mendeley Data, V1, doi: 10.17632/zr8tnvk43m.1

Point 3 – Measures of relationship quality

It would be interesting to see measures of relationship quality beyond the perception questionnaire. For example, how do financial management behaviors correlate with the probability of divorce? With the likelihood of having children? Tracing the participants for a follow-up survey could answer these questions and test the correlations over time.

Response: Thank you very much for this comment, it is an interesting and enriching remark. We will certainly consider in future research. Certainly, the limitation of the study of couples is the fact that they are declarations, self-reports. However, it would be worth considering the changes over the years. Planning longitudinal studies would also be valuable from our point of view.

Point 4 – Sample

Five hundred couples answered the survey, but how many were contacted and didn't complete it? Even if it's a representative sample demographically, attrition could create a biased sample, making it useful to report.

Response: Thank you for you valuable comment. In order to avoid misunderstandings we added the paragraph explaining the limitations of our sample. Representativeness could not be obtained in the study because our sample was not random. In selecting the sample, we started with the sample provided by the Central Statistical Office, which reflects a distribution of main characteristics of Polish population in terms of age / gender / voivodeship / town size. Our respondents took part in the Online Survey via the CAWI panel of the DRB Polonia company. Company controlled the distribution of those characteristics in selected sample.

„Due to the specific population of respondents we were not able to establish a sampling frame from which a random sample could be drawn. Consequently, our sample is pursposive not random. Although the purposive sampling does not yield the representativeness of results that could be projected to the population of dating couples in Poland, there are research showing that the non-probabilistic methods have similar levels of accuracy as probabilistic ones (Martinsson et al., 2013).”

Point 5 – Imputation

"The weighted predictive mean matching (midastouch) returned the lowest coefficient of variations." It is not clear to me that imputations should minimize variations. Instead, a good way to test the performance of the imputation models is to use some of the non-missing data as a test group. That is, say there are 100 complete observations (I didn't find the number of observations with missing values?), use 80 to train the model to predict each of the missing variables, then see what it predicts for the 20 observations left out. Finally, evaluate the model's accuracy (AUC/ROC for continuous, precision/recall for discrete) by comparing the test prediction with the actual observed values. 

Response: Thank you for this comment and suggestion. We will probably turn to AI methods when imputing data in the future, considering all pros and cons of ML.

The coefficient of variation measure has been chosen as a comparison method to preserve the joint and marginal distributions for both original and imputed sample. However, this measure is not well-recognized in the literature therefore we decided to apply the RMSE (root mean square error) to evaluate our models. We had missing data in 8 variables (8% of all answers).

Here are the results of RMSE for observed vs. imputed data for 8 variables.

 V1 V2 V3 V4 V5 V6 V7 V8 AVG. RMSE

Midas 5,051 5,287 9,433 9,118 12,234 12,805 5,567 5,614 8,139

Rforest 5,026 5,205 9,349 9,160 12,212 12,989 5,637 5,666 8,156

POLR 5,133 5,391 10,476 9,696 15,291 15,272 5,570 7,271 9,263

We did not insert this result to the manuscript due to the length of the article limitations. However, we decided to include this table in the answer to show that the imputation was done accurately and in accordance with the current state of knowledge in the field. 

Point 9 – Analyses explanation

For a non-APIM expert audience, the analysis is tough to follow. I think there should be some explanation (a paragraph?) of the models' math/statistics and intuition. For example, it's unclear why only these five variables were included in the models (especially the interaction between well-being and shared values). What was the test for which you reported the chi-square comparing exactly (the parameters of the two models?)? What are k and the other model parameters you refer to?

Response: Thank you for your comment. We agree that a few extra sentences about the model will make the text more reader-friendly. We added the following part:

“To meet this goal we specified a path model in which quality of relationship reported by male and female partner were predicted simultaneously by each actor’s and hers/his partner’s: KDM, hared Goals and Values, Harsh Startup; four financial management behavior variables: Cash Management, Savings and Investment, Credit Management, and Insurance. (…) Important question regarding this type of models is whether actors within a dyad may be considered indistinguishable, in our case with respect to gender. That means, whether determinants of well-being are the same, and their effects are of the same size, for both men and women. To test this assumption, we estimated two variants of our model: in one we imposed constraints on model parameters to hold values of analogous regression coefficients in the equation for men and the equation for women the same. In other one we enabled parameters to vary between actors of different gender.”

Point 10 – Descriptive analysis

It'd be helpful to see summary statistics about each variable used in the model and their intercorrelations. That will allow understanding of collinearities and interpretation of the main results (e.g., is an effect of 1.8 large?).

Response: Thank you. We added two correlations matrices to Appendix. We constructed two correlation matrices for Men and Women. However, we also tested our models with a Variance Inflation Factor (VIF) to make sure there is no multicollinearity. Here are score for models: Men, Women. None of the variables go beyond level 5, which means that the problem of collinearity does not occur here.

This is what we added to Appendix:

VIF results for two models (Men, Women) 

 KDM_M SGV_M Save_Inv_M Harsh_M Insurance_M Credit_Man_M Cash_Man_M 

 1.027307 1.796278 1.910548 1.264026 1.528631 1.199873 2.019205 

 KDM_K SGV_K Save_Inv_K Harsh_K Insurance_K Credit_Man_K Cash_Man_K 

 1.029460 1.394820 1.453528 1.347075 1.409441 1.205404 1.234295 

Corellation matrices for two models (Men, Women)

 KDM_M SGV_M Save_Inv_M Harsh_M Insurance_M Credit_Man_M Cash_Man_M

 KDM_M 1.00 -0.07 0.08 0.09 -0.01 0.07 0.00

 SGV_M -0.07 1.00 0.23 -0.43 0.30 0.15 0.59

 Save_Inv_M 0.08 0.23 1.00 -0.24 0.55 0.39 0.52

 Harsh_M 0.09 -0.43 -0.24 1.00 -0.21 -0.08 -0.29

 Insurance_M -0.01 0.30 0.55 -0.21 1.00 0.30 0.40

 Credit_Man_M 0.07 0.15 0.39 -0.08 0.30 1.00 0.24

 Cash_Man_M 0.00 0.59 0.52 -0.29 0.40 0.24 1.00

 KDM_K SGV_K Save_Inv_K Harsh_K Insurance_K Credit_Man_K Cash_Man_K

 KDM_K 1.00 -0.08 0.05 0.08 -0.01 0.06 0.10

 SGV_K -0.08 1.00 0.14 -0.47 0.28 0.07 0.23

 Save_Inv_K 0.05 0.14 1.00 -0.22 0.44 0.34 0.36

 Harsh_K 0.08 -0.47 -0.22 1.00 -0.18 0.02 -0.15

 Insurance_K -0.01 0.28 0.44 -0.18 1.00 0.32 0.30

 Credit_Man_K 0.06 0.07 0.34 0.02 0.32 1.00 0.20

 Cash_Man_K 0.10 0.23 0.36 -0.15 0.30 0.20 1.00

Thank you for that comment. We added the following 

Point 11 – Main results

Suppose the model's assumptions and parameters are accurate. The results show that both partners report higher relationship quality when women's cash management questionnaire scores increase. When women's insurance and shared goals and values questionnaire scores increase, their male partner weakly reports lower relationship quality. The effect of the other 20 variables was indistinguishable from zero. Also, I didn't understand how the interactional values arguments (lines 339-340) relate to the discussion, nor the size of the effect and its significance reported.

Also, I think Table 2 should include the number of observations 

Response: Thank you for a comment. As wrote in the beginning, the whole model mode rebuilt. Indeed, lines 339-340 were redundant, therefore we removed them from manuscript.

Point 12 – Conclusions

"Our results, seen from this perspective, may suggest that couples in Poland appreciate a division of financial labour." It might be better to ask that directly in the questionnaire because this does not seem to be a plausible interpretation of the positive parameters for the woman's cash management (most other financial parameters were statistically equal to zero).

"… whereby women deal with current financial affairs, while men are authorized to handle strategic ones." what evidence in the data supports that claim? The only related evidence provided is that when women score higher on the cash management questionnaire (for which I didn't see the questions), both partners report higher marriage quality.

*A few typos I found:

Line 27: "… man's cash management predicts changes in her assessment…"

Line 62: "… pursued hereDo…"

Line 67: "Second, we explore this link using dyadic data, which creates an opportunity to compare reports of both partners.. Fourth,"

Response: Thank you very much for your comments, the text of the article has been corrected.

---

## [Decision Letter · Decision Letter 2]

1 Dec 2022

In the quest for effective factors of satisfaction with life: Insights from intra-couple interaction and financial management variables

PONE-D-21-25538R2

Dear Dr. Baryła-Matejczuk,

We’re pleased to inform you that your manuscript has been judged scientifically suitable for publication and will be formally accepted for publication once it meets all outstanding technical requirements.

Kind regards,

José Alberto Molina

Academic Editor

PLOS ONE

Additional Editor Comments (optional):

Reviewers' comments:

Reviewer's Responses to Questions

**Comments to the Author**

1. If the authors have adequately addressed your comments raised in a previous round of review and you feel that this manuscript is now acceptable for publication, you may indicate that here to bypass the “Comments to the Author” section, enter your conflict of interest statement in the “Confidential to Editor” section, and submit your "Accept" recommendation.

Reviewer #3: All comments have been addressed

2. Is the manuscript technically sound, and do the data support the conclusions?

Reviewer #3: Yes

3. Has the statistical analysis been performed appropriately and rigorously? 

Reviewer #3: Yes

4. Have the authors made all data underlying the findings in their manuscript fully available?

Reviewer #3: Yes

5. Is the manuscript presented in an intelligible fashion and written in standard English?

Reviewer #3: Yes

6. Review Comments to the Author

Reviewer #3: The authors have done a great job refining their empirical analysis and presenting their data and results. I especially appreciated the conciseness and transparency of the model and discussion. I find the combination of the authors' survey with the APIM model convincing in describing the associations between reported well-being and financial management, which contributes to the literature on the topic. However, as acknowledged by the authors, the evidence does not imply that if one were to change their (or their spouse's) financial management, it would lead to higher reported well-being. Therefore, I think the second part of the title (i.e., starting after the ":") alone describes the manuscript more accurately.

7. PLOS authors have the option to publish the peer review history of their article (what does this mean?). If published, this will include your full peer review and any attached files.

Reviewer #3: No

---

## [Editor Report · Acceptance letter]

15 Dec 2022

PONE-D-21-25538R2 

In the quest for effective factors of satisfaction with life: Insights from intra-couple interaction and financial management variables 

Dear Dr. Baryła-Matejczuk:

I'm pleased to inform you that your manuscript has been deemed suitable for publication in PLOS ONE. Congratulations! Your manuscript is now with our production department. 

Kind regards, 

on behalf of

Professor José Alberto Molina 

Academic Editor

PLOS ONE